# Novel Knowledge of Macrolide Resistance in *Mycoplasma pneumoniae* by Azithromycin Exposure

**DOI:** 10.3390/microorganisms12010218

**Published:** 2024-01-21

**Authors:** Tomohiro Oishi, Nemu Hattori, Daisuke Yoshioka

**Affiliations:** 1Department of Clinical Infectious Diseases, Kawasaki Medical School, 577, Matsushima, Kurashiki 701-0192, Japan; 2Kawasaki Medical School, 577, Matsushima, Kurashiki 701-0192, Japan

**Keywords:** *Mycoplasma pneumoniae*, macrolide, mutation, azithromycin

## Abstract

The rise of macrolide-resistant *Mycoplasma pneumoniae* (MRMP), marked by point mutations in the 23S rRNA gene, poses a growing global concern since its initial detection in 2001. The prominence of the A2063G mutation during this emergence remains unexplained. This study aimed to clarify the possibility of detecting MRMP from recent clinical macrolide-susceptible *M. pneumoniae* through exposure to azithromycin (AZM), which has a long half-life and was launched immediately before the first MRMP detection. Six strains isolated from Japanese children in 2019 and reference strain (FH), all belonging to the recent dominant P1 genotype, two, or two subtype, were cultivated in a medium containing slightly higher concentrations than the originated minimum inhibitory concentration (MIC) of AZM and underwent sequencing if they grew. Four out of the seven strains grew after exposure to AZM, and C2617G and C2617A were detected, with no mutation in two strains. After another cultivation and sequencing, two of four strains grew, one was changed from C2617G to A2063G, and the other remained C2617A. The MIC of AZM in A2063G strains was 128 mg/mL; for C2617A, it was 0.0156 mg/mL. This is the first study to detect the strains with A2063G mutation from recent macrolide-susceptible *M. pneumoniae* using AZM exposure.

## 1. Introduction

*Mycoplasma pneumoniae* is the causative agent in community-acquired pneumonia, especially in children and young adults [1]. Macrolides are the first-line treatment for respiratory tract infections caused by *M. pneumoniae* [2]. However, macrolide-resistant *M. pneumoniae* (MRMP) was detected for the first time in 2001 in Japan [3]. The MRMP rate has increased in many countries, especially in Asia [4]. The mechanism of macrolide resistance is the point mutation in domain V of the 23S rRNA sequence, and positions 2063, 2064, and 2067 are the main mutation sites [5]. Among these, the A2063G transition is the most common, with a high resistance level to 14- and 15-membered macrolides, such as erythromycin (ERY), clarithromycin (CLR), and azithromycin (AZM) [5]. Though the reason MRMP had not been detected until 2000 remains unclear, it was more than 40 years after the first macrolide agent, ERY, was launched.

We have previously performed and have continued to conduct multicenter collaborative epidemiological studies on *M. pneumoniae* infections since 2008 [6,7]. Furthermore, we reported that the MRMP rate has been decreasing and that P1 genotypes (type 1 and type 2 and its subtypes), which consist of the P1 protein, important for *M. pneumoniae* to bind to the host epithelial cells, have changed regularly in their dominance over approximately 10 years [8], and the dominant genotype has recently changed from type 1 to type 2 and its subtypes [9] in Japan.

Thus, we hypothesized that AZM, launched in 2000, may be related to the occurrence of MRMP because of the short-term launches to detect MRMP and a long half-life. Thus, this study aimed to analyze more recent strains after the P1 dominant genotype in Japan to prevent the MRMP rate from increasing in the future. To the best of our knowledge, this is the first study addressing the isolated occurrence of MRMP through exposure to low-concentration AZM.

## 2. Materials and Methods

### 2.1. Ethical Aspects

The study protocol was approved by the Ethics Committee of Kawasaki Medical School, Kurashiki, Japan, on 8 September 2021 (no. 3119-05).

### 2.2. Sample Collection

*M. pneumoniae* samples used in this study were collected from pediatric patients with acute respiratory tract infections from 74 institutions located in eight areas across Japan (20 institutions in Kyushu, 25 in Chugoku, 3 in Shikoku, 11 in Kinki, 7 in Chubu, 3 in Kanto, 2 in Tohoku, and 3 in Hokkaido) from 2008, before the MRMP pandemic in Japan.

### 2.3. Strains

Among our collected samples, seven macrolide-susceptible *M. pneumoniae* were selected ensuring that they had no point mutation related to macrolide-resistance and their minimum inhibitory concentrations (MICs) were susceptible with microdilution methods [10], including six strains of P1 type 2 or 2g2 isolated in 2019 and FH of the standard strain (Table 1).

The six clinical strains were selected because the MRMP rate has been decreasing recently, and analyzing more recent isolates is deemed valuable for anticipating the potential re-emergence of MRMP. In addition, there has been a recent shift in the trends of the P1 types from type 1 to type 2 or its subtypes. Therefore, type 2 or 2g2 is considered suitable for understanding the current trend.

### 2.4. Laboratory Tests and Statistical Analysis

These seven strains were obtained by cultivating specimens. The medium used for isolation and determination of the MIC was pleuropneumonia-like organism broth (PPLO; Oxoid, Hampshire, UK) supplemented with 0.5% glucose (FUJIFILM Wako Pure Chemical Corporation, Osaka, Japan), 20% mycoplasma supplement G (Oxoid), and 0.0025% phenol red (Sigma-Aldrich, St. Louis, MO, USA).

The MICs of AZM (LKT Labs, Inc., Shenzhen, China) for these strains had already been determined with microdilution methods [10]. First, a medium containing 10^5^ to 10^6^ CFU/mL of *M. pneumoniae* was added to 96-well microplates and incubated at 37 °C for 6–8 days. The MIC was defined as the lowest concentration of the antimicrobial agent, wherein the metabolism of the organism was inhibited, evidenced by the lack of a color change in the medium 3 days after the drug-free control first showed a color change.

Next, exposures of low-concentration AZM were performed as follows. As the first exposure, the PPLO broth including AZM at concentrations of 0.002–1.6 mg/mL, which had higher concentrations than these seven strain MICs of AZM, was initially arranged. Then, these seven strains were cultivated in the PPLO broth including AZM at concentrations of 0.002–1.6 mg/mL for 28 days. When strains were present after cultivations, these were identified in the mutation sites (2063, 2064, and 2617) in domain V of the 23S rRNA of *M. pneumoniae* using a direct sequencing method [7].

We performed the second exposure of AZM at higher concentrations than the ones that were able to be cultivated during the first AZM exposure for 28 days, and the strains that were able to be cultivated after the second AZM exposure were performed using direct sequencing at sites 2063, 2064, and 2617 in domain V of the 23S rRNA of *M. pneumoniae* The MICs of the isolates, which detected the mutations by the direct sequencing after exposures of low-concentration AZM, were determined the same way [10].

## 3. Results

Four out of these seven strains grew after the first AZM exposure in mediums including AZM 0.002–1.6 mg/mL Mutations were not detected in two of those; but, in the other two strains (M1601 and M1613), we detected C2617G and C2617A mutations, respectively (Table 2).

These four strains were cultivated in mediums including AZM 0.16–128 mg/mL as the second exposure to low concentration. Two out of these four strains grew after the second AZM exposure. Mutations were not detected in two of those after the second AZM exposure did not grow, but the other two strains (M1601 and M1613) grew until the medium included 128 and 16 mg/mL of AZM, respectively. These two strains were sequenced, and in M1601, we detected the A2063G mutation, which was changed before the second AZM exposure. However, in M1613, we detected the C2617A mutation, which was not changed before the second AZM exposure (Table 3).

These two strains after the second AZM exposure determined the MIC of AZM. The MIC of AZM in M1601 and M1613 was 128 and 0.0156 mg/mL, respectively (Table 4).

## 4. Discussion

Two previous studies reported that to induce macrolide resistance in *M. pneumoniae* in vitro, exposure to macrolide agents was needed [3,11]. For example, clinical samples were used but only for ERY as the exposure macrolide agent [3] Furthermore, the clinical samples used in their study were isolated more than 20 years ago [3]. The other report only used the reference strain of *M. pneumoniae* [11]. In addition, Okazaki et al. mentioned that 7.8% of the EM-sensitive isolates were detected in point mutations in the 23S rRNA of *M. pneumoniae* after grew in the medium containing 100 μg/mL of EM during incubation for 10–28 days. The mutations detected after growth were A2063G, A2064G, and A2064C. Pereyre et.al. reported that in the macrolide-susceptible reference strain M129 of *M. pneumoniae* after 23–50 serial passages in subinhibitory concentrations of some kinds of macrolides, they detected C2611A mutation by ERY and AZM and A2063G mutation by Josamycin. Comparing our findings to these prior reports, our study aligns with the report by Okazaki et al. in terms of the detection period of mutations, which was much shorter than that in Pereyre’s report. This difference may be attributed to the possibility of quasispecies in *M. pneumoniae*. Quasispecies, commonly used to describe sequence variants in heterogeneous virus populations, in the case of clinical isolates of *M. pneumoniae*, means they comprise mixed populations of drug-sensitive and drug-resistant molecular mutants. Chan et al. mentioned that in 48.2% of the clinical samples of *M. pneumoniae,* they detected quasispecies using pyrosequencing [12]. Thus, it is suggested that some clinical strains of *M. pneumoniae*, which were diagnosed as macrolide-sensitive, may already contain populations of macrolide-sensitive and macrolide-resistant molecular mutants. Consequently, the detection periods might be much shorter among clinical isolates than those in the reference strain.

Naturally, bacteria are known to have certain mutation rates, and mutations related to antibiotic resistance are no exception [13]. Therefore, it is possible that macrolide-resistant strains arise in isolates that already have populations of both macrolide-sensitive and macrolide-resistant molecular mutants after exposure to macrolides. However, MRMP was first detected in Japan in 2001 [3] marking the first instance since the launch of the first macrolide agent, EM, in 1955.

Specifically, AZM exhibits a longer half-life and greater distribution into tissues and fluids compared to other macrolides [14,15]. This characteristic suggests that AZM can persist in the body at low concentrations for an extended duration, aligning with our study protocol. Moreover, it appears not to be coincidental that AZM was launched in 2000, immediately before the appearance of MRMP for the first time in 2001. A2063G mutation is the most prevalent point mutation associated with macrolide resistance in *M. pneumoniae*, as previously mentioned [9]. This may be because, among other mutations, including C2617A or C2617G detected in our study, their MICs of AZM are much lower than that of A2063G. Consequently, they are more susceptible to being eliminated by macrolide antibiotics, even at relatively lower concentrations.

Another novel aspect in our study was the detection of mutations related to macrolide resistance in recent clinical isolates of *M. pneumoniae*. As previously mentioned, the MRMP rate has been decreasing recently, and the major P1 genotype has shifted from type 1 to type 2 and its subtypes [9]. Since it has not been long since these isolates appeared and became predominant, they have had fewer opportunities for exposure to macrolide agents. Despite the rates of MRMP being much lower among strains of type 2 and its subtypes compared to type 1 [9], there is a possibility that these isolates acquired point mutations related to macrolide resistance through exposure to AZM. Therefore, our results serve as a reminder to exercise caution in antibiotic use to prevent the increase in MRMP.

This study had some limitations. First, only one type of macrolide agent was used. While using other macrolides could be considered, we chose AZM due to its long half-life, making it more likely to induce mutations upon exposure in vivo. Second, the sample size was small, raising uncertainty about how frequently mutations arise in macrolide-susceptible *M. pneumoniae* upon exposure to AZM. Thus, future studies should include a larger number of strains to address this limitation. Finally, there was uncertainty regarding whether the parent isolates used were originally a mix of macrolide-susceptible *M. pneumoniae* and MRMP mutants. Confirming this would require specialized methods, such as pyrosequencing [12].

Nevertheless, the crucial point remains that even recent clinical samples of macrolide-susceptible *M. pneumoniae* are prone to transition into MRMP, including the A2063G mutant with a high resistance mutation, upon exposure to AZM. It is important to note that these limitations do not compromise the conclusions drawn from this study.

## 5. Conclusions

Our study is the first to identify strains with the dominant A2063G mutation from recent macrolide-susceptible *M. pneumoniae* using AZM exposure. This highlights the potential emergence of MRMP with the use of macrolides, such as AZM, emphasizing the need for careful antibiotic management.

## Figures and Tables

**Table 1 microorganisms-12-00218-t001:** Characteristics of macrolide-sensitive Mycoplasma pneumoniae.

Strain No.	Isolation Date	Mutations	MICs of AZM (μg/mL)	P1 Type
FH (standard strains)	-	-	0.001	2
M1601	Jul-2019	-	0.0005	2
M1603	Aug-2019	-	0.001	2g2
M1611	Nov-2019	-	0.001	2g2
M1613	Nov-2019	-	0.001	2g2
M1634	Jun-2020	-	0.00012	2g2
M1644	Sep-2020	-	0.0005	2g2

**Table 2 microorganisms-12-00218-t002:** Changes in the mutations of *Mycoplasma pneumoniae* after first exposure to low-concentration azithromycin *.

Strain No.	Maximal Concentrations (Strains Grew)	Mutations	Original MICs of AZM (μg/mL)
FH (standard strains)	No growth	Not possible	0.001
M1601	0.004	C2617G	0.0005
M1603	0.004	-	0.001
M1611	0.004	-	0.001
M1613	0.004	C2617A	0.001
M1634	No growth	Not possible	0.00012
M1644	No growth	Not possible	0.0005

AZM: azithromycin. MIC: minimum inhibitory concentrations. * Exposure of low-concentration azithromycin: Exposure with the PPLO broth including AZM at the concentrations from 0.002 to 1.6 g/mL, higher concentrations than the original MICs of AZM.

**Table 3 microorganisms-12-00218-t003:** Changes in the mutations of Mycoplasma pneumoniae after the second exposure of low-concentration azithromycin.

Strain No.	Maximal Concentrations (Strains Grew)	Mutations	Last Mutations
M1601	128	A2063G	C2617G
M1603	No growth	Not possible	-
M1611	No growth	Not possible	-
M1613	16	C2617A	C2617A

**Table 4 microorganisms-12-00218-t004:** MICs of azithromycin for Mycoplasma pneumoniae after the second exposure to low-concentration azithromycin.

Strain No.	MICs of AZM (μg/mL)	Mutations	Original MIC (μg/mL)
M1601	128	A2063G	0.0005
M1613	0.0156	C2617A	0.001

AZM: azithromycin. MIC: minimum inhibitory concentrations.

## Data Availability

The data that support the findings of this study are available from the corresponding author, Tomohiro Oishi, upon reasonable request.

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
