# Peer review of "Novel Knowledge of Macrolide Resistance in Mycoplasma pneumoniae by Azithromycin Exposure"

_microorganisms, 2024, doi:10.3390/microorganisms12010218_

Round 1

Reviewer 1 Report

Comments and Suggestions for Authors

In the study, Novel Knowledge of Macrolide Resistance in Mycoplasma pneumoniae by Azithromycin Exposure", the authors present very preliminary and limited data about the detection of certain point mutations associated with exposure to low concentrations of one macrolide. 

- The study depended on a very low number of isolates

- The isolates investigated were not verified to ensure they are not a mixed population of sensitive and resistant cells.

- The study did not confirm the relationship between the detected mutations and observed change in MIC, for example by specifically inducing the change or complementing the obtained mutants, if possible.  

Comments on the Quality of English Language

No comments

Reviewer 2 Report

Comments and Suggestions for Authors

This study investigates the mechanism of action of Mycoplasma pneumoniae resistance to azithromycin.  

Although this study follows the procedures of previous investigations, has two original things, it used new strains isolated from children and used azithromycin which is the macrolide of greater persistence in human organisms.  

This research brings as a novelty the two points cited above.    The methodology used is molecular biology techniques already employed in other studies.    

The conclusions are links to the evidence of this investigation and are adequate. As I mentioned earlier the number of strains is small, but this study opens new paths for future research.

Reviewer 3 Report

Comments and Suggestions for Authors

In the MS microorganisms-2838479, the authors report identifying strains with the dominant A2063G mutation (the most prevalent point mutation associated with macrolide resistance in M. pneumoniae) in a recent macrolide-susceptible M. pneumoniae strain exposed to Azithromycin. They associate this phenomenon with the particular pharmacokinetic profile of Azythromicin compared to other macrolide representatives: a longer half-life and higher distribution into tissues and fluids.

Their original study is well-designed, the methods are well-described and the results are displayed with corresponding details. The authors are objective, pointing out potential limitations, but the results and discussion are relevant enough to support the conclusions. 

Overall, the present communication has substantial importance and application in clinical practice, evidencing the potential occurrence of macrolide-resistant M. pneumoniae strains during the treatment with Azithromycin and, therefore, claiming again the need for careful management of antibiotic use.

Reviewer 4 Report

Comments and Suggestions for Authors

The research proposed by the authors is interesting, and the evaluation was based on:

1.     The introduction provides very systematized information about the pathogen Mycoplasma pneumoniae which is responsible for the emergence of complex pathologies in the respiratory system; data on macrolide antibiotics, with targeted reference to azithromycin, as well as the emergence of multiple forms of resistance; the purpose of the research is well defined;

2.     The material and methods chapter, I think is described properly; the authors describe for each sub-stage of the research the type of reagents, the research methods used as well as the study protocol approved by the Ethics Commission; the work protocol is presented in detail; samples collected from patients from more than 8 areas in Japan were used;

3.     Results and discussions are presented on each sub-stage of the research; the obtained results are correlated with the data from the specialized literature;

4.     The conclusions are consistent with the proposed purpose and potential premises for further research;

5.     I consider the bibliography to be justifiable.

It is worth mentioning the originality of the research and the multiple signals regarding the establishment of antibiotic resistance of different infectious agents, including Mycoplasma pneumoniae.